# The impact of COVID-19 on the physical activity and sedentary behaviour levels of pregnant women with gestational diabetes

**Medbh Hillyard**[1], **Marlene Sinclair**[1]*, **Marie Murphy**[2], **Karen Casson**[1], **Ciara Mulligan**[3]

**1** Institute of Nursing and Health Research, Ulster University, Newtownabbey, Northern Ireland, **2** Sport and Exercise Sciences Research Institute, Ulster University, Newtownabbey, Northern Ireland, **3** Ulster Hospital, Dundonald, South Eastern Health and Social Services Trust, Newtownabbey, Northern Ireland

* m.sinclair1@ulster.ac.uk

## Abstract

### Background

The aim of this study was to understand how physical activity and sedentary behaviour levels of pregnant women with gestational diabetes in the UK have been affected by COVID-19.

### Methods

An online survey exploring physical activity and sedentary behaviour levels of pregnant women with gestational diabetes during COVID-19 was distributed through social media platforms. Women who had been pregnant during the COVID-19 outbreak and had gestational diabetes, were resident in the UK, were 18 years old or over and could understand written English were invited to take part.

### Results

A total of 724 women accessed the survey, 553 of these met the eligibility criteria and took part in the survey. Sedentary time increased for 79% of the women during the pandemic. Almost half of the women (47%) were meeting the physical activity guidelines pre COVID-19 during their pregnancy, this dropped to 23% during the COVID-19 pandemic. Fear of leaving the house due to COVID-19 was the most commonly reported reason for the decline. Significant associations were found between meeting the physical activity guidelines during COVID-19 and educational attainment, fitness equipment ownership and knowledge of how to exercise safely in pregnancy.

### Conclusions and implications

These results show the impact of COVID-19 on physical activity and sedentary behaviour levels and highlight the need for targeted public health initiatives as the pandemic continues and for future lockdowns. Women with gestational diabetes need to know how it is safe and beneficial to them to engage in physical activity and ways to do this from their homes if fear

**Data Availability Statement:** Data are available from Ulster University. https://doi.org/10.21251/63d59840-e88b-43b8-a0da-74a152162f1f.

**Funding:** MH was supported by a Studentship funded by the Department for the Economy, Northern Ireland. The funder had no role in study design, data collection and analysis, decision to publish, or preparation of the manuscript.

**Competing interests:** The authors have declared that no competing interests exist.

of leaving the house due to COVID-19 is a barrier for them. Online physical activity classes provided by certified trainers in physical activity for pregnant women may help them remain active when face-to-face appointments are reduced and limited additional resources are available.

## 1. Background

The outbreak of COVID-19 has had a huge impact on all areas of society; changing working patterns, restricting movement and social interactions, and increasing caring responsibilities and home schooling. For pregnant women it resulted in reduced face-to-face antenatal appointments, including exposure to general health information/advice routinely displayed in clinics and discussions with staff and peers. Pregnant women have experienced increased fear and anxiety about catching the virus whilst pregnant in case of harm to their baby and themselves [1]. Pregnancy is a precious time in a woman's life during which she may feel more vulnerable.

In the UK, pregnant women were placed in the vulnerable category on the 16th March 2020, and, therefore, advised to reduce social contact through social distancing [2]. The current available evidence indicates that pregnant women with pre-existing comorbidities, high maternal age and high body mass index who contract COVID-19 may be more likely to be admitted to an intensive care unit and preterm birth rates are higher in pregnant women with COVID-19 than in pregnant women without the virus [3].

Although social distancing and shielding are thought to impact an individual's physical activity and sedentary behaviour levels there are conflicting reports about the magnitude and direction of this impact. It is thought that some individuals may have more time for structured exercise, however, their sedentary time may have increased due to working from home and having lost the daily physical activity associated with personal transport and incidental physical activity associated with their usual working environment and practices [4]. However, the overall picture of physical activity levels is unclear, especially for pregnant women. The benefits of physical activity (PA) during pregnancy are well-established. Being active has been associated with a reduction in the occurrence of gestational diabetes [5–7], gestational hypertension disorders [8], macrosomia [9, 10], and excess weight gain [11], as well as shorter labours [12] and improved mood [13].

In June 2017, the UK's four Chief Medical Officers released new physical activity guidance for pregnancy, recommending pregnant women should aim to do 150 minutes of moderate intensity physical activity each week, the same as their non-pregnant counterparts. Despite the evidence, only 51% of non-pregnant women in Northern Ireland undertook sufficient physical activity for optimum health pre-Coronavirus [14]. In 2019, new UK-PA guidelines included this recommendation for pregnant women, highlighting the importance and benefits of pregnant women being active [15]. The body of literature on which these recommendations are based, found no evidence of harm for the mother or infant as a result of 150 minutes of moderate intensity physical activity per week. However, PA levels of women naturally decrease during pregnancy [16–19], due to factors such as tiredness, sickness and pain related to pregnancy [20].

As the general population's PA has changed during the pandemic, it is likely that pregnant women's PA levels may have also changed. Commonly reported physical activities in

pregnancy include face-to-face antenatal classes such as pregnancy yoga, Pilates and swimming [21], none which have been possible during the pandemic.

Pregnant women have faced many challenges during the COVID-19 pandemic, and women diagnosed with conditions such as gestational diabetes mellitus (GDM) are likely to have faced additional challenges and increased concern. GDM is glucose intolerance, which begins or is first diagnosed during pregnancy [22]. Risk factors for GDM include being overweight or obese [23]; high maternal age [24]; having a first degree relative with diabetes; previous pregnancy with GDM [25]; being of South Asian origin [26]; being of Black Caribbean origin [27]; being of Middle Eastern origin [28] and being sedentary or inactive [29]. Women diagnosed with GDM are often advised in the first instance to try and control their glucose levels through diet and physical activity [30].

With COVID-19 restrictions the normal testing procedures and care pathways have had to change to limit women's face-to-face contact with clinicians. The RCOG guidelines changed during COVID-19 and no longer recommended women at risk from gestational diabetes undergo an Oral Glucose Tolerance Test (OGTT) due to the risk of having to sit in the hospital for a prolonged period of time. At the height of the pandemic women were being diagnosed through HbA1c tests at various time points [2]. The latest guidelines suggest a flexible approach to screening and maternity care is needed to respond to changes in risk at a local or national level [2]. Due to COVID-19, once women are diagnosed with GDM they are having a reduced number of face-to-face appointments, with a heavy reliance on remote communication. It is unclear what impact this will have had on their pregnancy.

Physical activity in pregnant women with GDM is particularly important due to the potential to improve blood glucose control [31] and reduce the need for medication [32]. A meta-analysis on the effect of physical activity on maternal and fetal outcomes in women with GDM found that women in the PA intervention groups were 47% less likely to require insulin, compared to those in the control groups (OR 0.53, 95% CI 0.29, 0.97, P = 0.04) [32]. Given the vital importance of PA to pregnant women with GDM, the NICE guidelines recommend pregnant women diagnosed with gestational diabetes are offered advice about changes in diet and exercise [30]. Therefore, understanding the impact of lockdown and social distancing on pregnant women's physical activity levels for women with gestational diabetes is particularly important.

Furthermore, a study of COVID-19 in pregnancy found that women who were over 35 years and overweight or obese were at greater risk of developing severe illness if they contract COVID-19, both of which are also risk factors for GDM [2]. Therefore, the aim of this study was to understand how COVID-19 has affected the self-reported physical activity and sedentary behaviour levels of pregnant women with GDM. The findings of this study will help policy makers and health service providers to understand how best to support pregnant women during subsequent waves of COVID-19 or future pandemics or situations requiring lockdown.

This study was based on the COM-B model of behaviour which focuses on the capability, opportunity and motivation for health behaviours. These elements influence whether or not a behaviour will take place [33]. Each of these three domains can then be further split into two sub-domains. Capability focuses on aspects such as an individual's skills, strength and knowledge to choose a particular behaviour [33]. Opportunity includes elements such as social cues, cultural norms, time and the environment; this domain can be further divided into social and physical sub domains [33]. Motivation can be divided into reflective motivation, which is considered and usually involves a plan and automatic motivation which is driven by emotions and is often reactions to events [33].

## 2. Methods

### 2.1 Respondents

Women who had been pregnant during the COVID-19 outbreak and had gestational diabetes, were resident in the UK, were 18 years old or over and could understand written English were invited to take part.

### 2.2 Measures

The survey was developed on the Qualtrics platform. Once consent had been given the respondent was required to answer three screening questions to ensure eligibility; answering no to any of these questions resulted in the survey ending for the participant.

The survey included questions on:

- Demographics (e.g., age, employment status, qualifications, ethnicity, relationship status, parity)

- Individual's circumstances (e.g., living arrangements, access to various spaces for physical activity, ownership of gym equipment, wearing activity tracker, knowledge of PA in pregnancy)

- Health and pregnancy (e.g., due date, diagnosis of additional health conditions, GDM management)

- Activity levels (e.g., Single item PA measure [34])

- General Worry using the Brief Measure of Worry Severity scale [35]

**2.2.1 Physical activity.**    PA levels were assessed using the single-item physical activity measure [34]. Women were asked "In an average week, **prior** to the COVID-19 outbreak but during your pregnancy, on how many days would you have done a total of 30 minutes or more of physical activity, which was **enough** to raise your breathing rate?", they were also asked the same question with regards to their PA levels during the COVID-19 outbreak. Participants were then categorised into three groups; increased, decreased and no change. Based on the UK PA guidelines the women were also categorised into meeting the PA guidelines/not meeting the guidelines at both time points.

**2.2.2 Sedentary Behaviour (SB).**    Sedentary behaviour was assessed by asking the question "During the COVID-19 outbreak was your sedentary time much less than normal, a little less than normal, about the same, a little more than normal, a lot more than normal. For analysis, sedentary time was categorised as 'decreased', 'increased' and 'no change'.

**2.2.3 Worry.**    Worry was assessed using the Brief Measure of Worry Severity (BMWS) [35], an 8-item scale which asked the women about their general/usual experience of worrying, selecting one option for each question out of not true at all = 0, somewhat true = 1, moderately true = 2 and definitely true = 3. Scores for the eight questions were summed to give an overall worry score (range = 0 to 24). Cronbach's alpha was calculated and showed strong reliability for this scale (Cronbach's alpha, 0.93, 95% CI 0.92–0.94). For some analysis worry was dichotomised into low worry (0–12) or high worry (13–24).

**2.2.4 COM-B.**    Attitudes to physical activity, both before and during COVID-19 were assessed by asking for agreement with statements based on the capability, opportunity and motivation model of behaviour (COM-B) [33].

The development of the questionnaire was informed by the Sport England [36] and Active Pregnancy Foundation [37] COVID-19 surveys.

## 2.3 Procedures

The survey was live from Sunday 5[th] July until Monday 20[th] July 2020. It was shared online through a wide range of Facebook Groups, which included pregnancy and parenting groups, gestational diabetes groups, COVID-19 support groups and health and well-being groups. It was also shared widely through Twitter networks. The researcher contacted the administrators of the various social media pages to request permission to post on the sites where required. The social media post contained details of the study and who was eligible to take part, plus a link to the survey. Once interested participants clicked on the link, they were taken to a participant information sheet which provided full details about the study, they were also provided with links to support services. All data was collected in line with the terms and conditions of the websites, with appropriate permissions. The full questionnaire and data set are available online.

## 2.4 Sample size

There were 731,213 births in the UK in 2018 [38]. It is estimated that approximately 4.4% of women in the UK have GDM [30]. This would mean approximately 32,173 women are diagnosed with GDM annually in the UK. Based on a power calculation with a population size of 32,173, at a 95% confidence level and 5% margin of error, it was calculated that 380 women would need to complete the survey to provide representative results.

## 2.5 Data analysis

Data were analysed using SPSS Version 26. Means and standard deviations were calculated for continuous variables and frequencies and percentages for categorical variables. For the purpose of analysis some variables were turned into dichotomous variables; meeting the PA guidelines/not meeting PA guidelines, degree/no degree, GDM diet controlled/not diet controlled. Chi-square tests were used to explore the associations between demographics/individual's circumstances and meeting the PA guidelines. Odds ratios were calculated for various variables in relation to meeting the PA guidelines during COVID-19. A One-way ANOVA was used to compare the difference in mean worry scores between those meeting the PA guidelines during COVID-19 and those who did not meet the guidelines. A logistic binary regression model was used to explore the association between physical activity levels during COVID-19, key demographic variables and work/living arrangements during the pandemic and a logistic binary regression model was also used to investigate how the components of COM-B might influence PA levels.

## 2.6 Ethical considerations

Ethical approval was obtained from the University's Nursing and Health Research Ethics Filter Committee (Ref FCNUR-20-09b). Women considering taking part in the study were provided with full details about what involvement would entail through the participant information sheet which was displayed when they clicked on the link to the survey. Women were not able to begin the survey unless they had provided consent to take part by selecting the appropriate box. Women were provided with contact details for support sites available which contained information on; maternal mental health during pregnancy, information on COVID-19 during pregnancy and details of available support services. All responses were anonymous.

## 3. Results

A total of 724 women accessed the survey, 553 of these women were eligible to take part and completed all or some of the survey. Fig 1 shows the participant flow chart to achieve the study sample.

The mean age of participants was 32.1 years (SD: 4.7; range 19–44 years), 92% were in a relationship/married and living together, 93% were white and 59% of the women had an undergraduate degree or higher degree. The majority of respondents lived in England (74%) with smaller proportions from the other three UK countries. The characteristics of respondents are displayed in Table 1 and are also presented according to meeting the PA guidelines/not meeting PA guidelines during COVID-19.

### 3.1 Pregnancy

The mean gestation of the women in the sample was 22.1 (SD 8.2) weeks at the start of lockdown, with 13% of women in their first trimester, 57% in trimester two and 30% in trimester three. The majority of women were multiparous (62%). The mean number of weeks pregnant the women were at GDM diagnosis was 24.4 (SD 7.0) and 51% of the women managed the GDM through diet. Pregnancy and health statistics are displayed in Table 2.

### 3.2 Physical activity levels

The mean number of days women were doing 30 minutes or more of moderate intensity PA was 4.0 days (SD 2.9) before COVID-19 and 2.8 days (SD 2.2) during COVID-19. Women who were doing 30 minutes or more of moderate intensity PA on five or more days of the week were categorised as meeting the UK PA guidelines of 150 minutes of moderate intensity PA per week. This equates to 47% of the sample meeting the UK PA guidelines pre COVID-19 and 23% of women meeting the PA guidelines during COVID-19. Women in their first and third trimesters were less likely to meet the PA guidelines during COVID-19 than women in their second trimester (T1 13.1%, T2 28.2%, T3 18.2%, P = 0.008).

Overall, 60% of the women decreased their PA levels during COVID-19, 21% did not change their PA levels and 19% increased their PA levels.

**3.2.1 Impact of women's circumstances upon meeting PA guidelines during COVID-19.** Women were asked a number of questions around their working arrangements during COVID-19; were they on maternity leave, had they worked from home at all, were they a key worker? They were also asked questions on their personal circumstances; Do they have space at home to exercise, do they have any fitness equipment at home, do they know how to exercise safely in pregnancy, do they have other children? These factors, along with educational status (degree/no degree) were used in a logistic binary regression model to see what factors were possible predictors of meeting the PA guidelines during COVID-19. Significant factors for meeting the PA guidelines during COVID-19 were having an undergraduate or higher degree, having fitness equipment at home and knowing how to exercise safely in pregnancy (Table 3).

Women with high self-reported worry scores were less likely than women with low worry scores to meet the PA guidelines (OR 0.58, 95% CI 0.367–0.92). Women with reported long-term health conditions were less likely than women with no long-term health conditions to meet PA guidelines (OR 0.389, 95% CI 0.21–0.71).

### 3.3 Reasons for decline in physical activity levels

Those whose physical activity levels had declined during the pandemic were asked to select which factors contributed to the decline. The most commonly reported reason was fear of

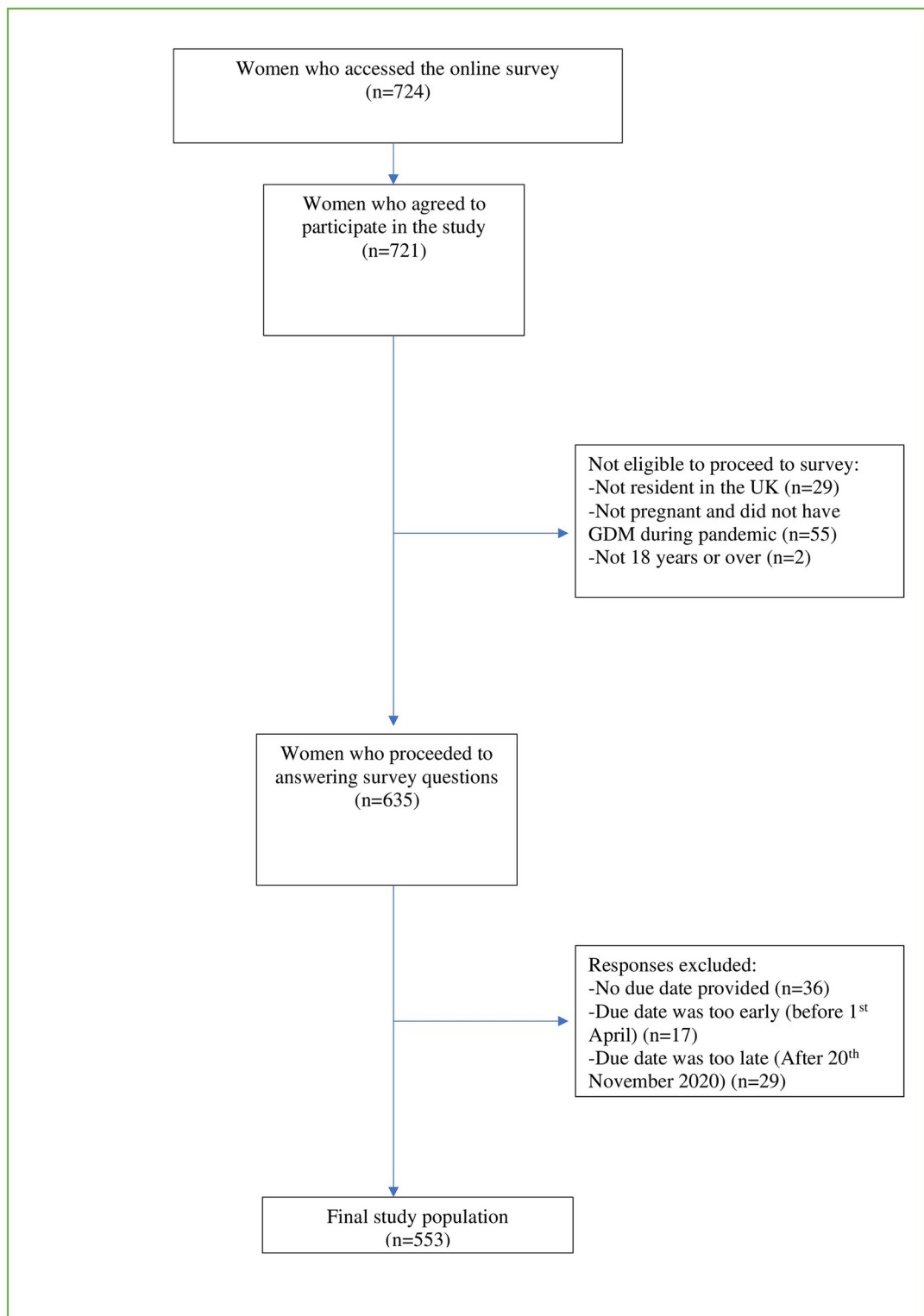

**Fig 1. Participant flow chart to achieve study sample.**

**Table 1. Characteristics of respondents from online survey for pregnant women with GDM during COVID-19 pandemic according to meeting the physical activity guidelines/not meeting the physical activity guidelines.**

| Maternal characteristics | | | | |
|---|---|---|---|---|
| Age (years) | All | Meeting PA guidelines during COVID-19 | Not meeting PA guidelines during COVID-19 | P Value |
| Mean (SD) | 32.1 (4.7) | 33.0 (4.3) | 31.8 (4.8) | .032* |
| | *No. (%)* | *No. (%)* | *No. (%)* | |
| *Educational qualifications* [a] | n = 419 | n = 103 | n = 315 | .004* |
| Undergraduate degree or higher | 246 (58.7) | 73 (29.7) | 173 (70.3) | |
| No degree | 173 (41.3) | 30 (17.4) | 142 (82.6) | |
| *Key worker* [a] | n = 418 | n = 104 | n = 313 | .929 |
| Yes (%) | 182 (43.5) | 45 (24.7) | 137 (75.3) | |
| No (%) | 236 (56.5) | 59 (25.1) | 176 (74.9) | |
| *Country* [a] | n = 413 | n = 102 | n = 310 | .561 |
| England (%) | 305 (73.8) | 77 (25.3) | 227 (74.7) | |
| Northern Ireland | 73 (17.7) | 19 (26) | 54 (74) | |
| Scotland | 30 (7.3) | 5 (16.7) | 25 (83.3) | |
| Wales** | 5 (1.2) | 1 (20) | 4 (80) | |

[a] Cell counts may not add up to total number of respondents due to missing values.

*Statistically significant result.

**England and Wales were combined for analysis due to small numbers for Wales.

**Table 2. Pregnancy and health characteristics of respondents from online survey for pregnant women with GDM during COVID-19 pandemic according to meeting the physical activity guidelines/not meeting physical activity guidelines.**

| Pregnancy and health characteristics | | | | |
|---|---|---|---|---|
| Gestation at start of lockdown (24.3.20) (weeks) | All | Meeting PA guidelines during COVID-19 | Not meeting PA guidelines during COVID-19 | P Value |
| Mean (SD) | 22.1 (8.2) | 22.0 (7.5) | 22.4 (8.4) | .779 |
| | *No. (%)* | *No. (%)* | *No. (%)* | |
| *Long term health conditions lasting longer than 12 months* [a] | n = 419 | n = 104 | n = 314 | .001* |
| Yes | 111 (26.5) | 15 (13.6) | 95 (86.4) | |
| No | 308 (73.5) | 89 (28.9) | 219 (71.7) | |
| *Other children* [a] | n = 416 | n = 102 | n = 313 | .985 |
| No (%) | 159 (38.2) | 39 (24.5) | 120 (75.5) | |
| Yes (%) | 257 (61.8) | 63 (24.6) | 193 (75.4) | |
| *Number of weeks pregnant at GDM diagnosis* | | | | .806 |
| Mean (SD) | 24.4 (7.0) | 24.2 (7.3) | 24.5 (6.8) | |
| *Treatment* [a] | n = 553 | n = 117 | n = 385 | .838 |
| Diet only (%) | 280 (50.6) | 64 (25) | 192 (75) | |
| Medication | 273 (49.4) | 53 (21.5) | 193 (78.5) | |

*Statistically significant result.

[a] Cell counts may not add up to total number of respondents due to missing values.

**Table 3. Personal circumstances as predictors of meeting the PA guidelines during COVID-19.**

| Predictor | B | SE | df | Sig. | Exp (B) | 95% CI for Exp (B) | |
|---|---|---|---|---|---|---|---|
| | | | | | | Lower | Upper |
| Education (degree/no degree) | .556 | .278 | 1 | .045* | 1.743 | 1.012 | 3.003 |
| Fitness equipment at home | .517 | .252 | 1 | .040* | 1.677 | 1.024 | 2.746 |
| Knowing how to exercise safely | .584 | .266 | 1 | .028* | 1.794 | 1.065 | 3.020 |
| Space to exercise | .440 | .288 | 1 | 1.27 | 1.553 | .882 | 2.733 |
| Key worker | .059 | .256 | 1 | .818 | 1.061 | .642 | 1.752 |
| Other children | .046 | .253 | 1 | .856 | 1.047 | .637 | 1.721 |
| Maternity leave | .259 | .258 | 1 | .317 | 1.295 | .780 | 2.150 |
| Employment status | .013 | .366 | 1 | .972 | 1.013 | .494 | 2.077 |
| Working from home | .209 | .310 | 1 | .501 | 1.232 | .671 | 2.262 |

*Statistically significant result.

B- co-efficient for the constant.

SE-Standard error.

df-degrees of freedom.

Sig.-Significance.

Exp (B) Exponentiation of B coefficient (odds ratio).

CI- confidence intervals.

leaving the house due to COVID-19 (69%). Other common reasons given for the decline were lack of motivation (58%), advancing stage of pregnancy, (56%) and lack of energy (51%).

There were some differences between subgroups with responses given to explain the decline in PA levels. Those with an undergraduate or higher degree were more likely to give the reasons their normal activities were not provided due to COVID-19 (62% Vs. 40%, $X^2$ = 11.657, P < .001) and working from home (52% Vs. 23%, $X^2$ = 20.713, P < .001). Those who had higher worry scores were more likely than those with lower worry scores to give the reasons 'found lockdown hard' (50% Vs. 25%, $X^2$ = 16.348, P < .001), 'lack of energy' (58% Vs 44%, $X^2$ = 4.542, P = .033) and 'lack of motivation' (72% Vs. 46%, $X^2$ = 17.538, P < .001).

### 3.4 Reasons for increasing PA levels

The most commonly reported reasons for doing more PA during COVID-19 were better weather (79%), felt it was important to be physically active due to GDM (62%) and exercise was an approved reason to leave the house (58%).

### 3.5 Knowledge around PA in pregnancy

Women reported their knowledge of PA in pregnancy by selecting their level of agreement with a number of statements on a 5-point Likert scale. For analysis purposes responses were recoded into two categories of agree and neither agree nor disagree and disagree. Just over six in ten women (62%) agreed that they knew how to exercise safely in pregnancy and 92% of women agreed it would be useful to receive information on the PA guidelines in pregnancy. Just over four in ten women (42%) of women in the study had sought resources to help them be physically active during lockdown. The most common sources for information were social media (56%) and websites (55%) and the least reported source was from health professionals (13%). Over three quarters of the women in the study (76%) agreed they would take part in an online pregnancy exercise class if it was available.

Table 4. Agreement with COM-B statements before and during COVID-19.

| | % Strongly Agree/Agree | % Strongly Agree/Agree | |
| --- | --- | --- | --- |
| | Before COVID-19 | During COVID-19 | P value |
| I had the ability to be physically active | 86.5% | 58.6% | < .001* |
| I had the opportunity to be physically active | 88% | 51.1% | 0.099 |
| It was important to me to be physically active | 75.1% | 65.9% | < .001* |
| I found exercise enjoyable and satisfying | 59.9% | 37.5% | < .001* |
| I felt guilty when I don't exercise | 51.9% | 56.9% | < .001* |

*Statistically significant result.

## 3.6 COM-B

The COM-B model was used to assess women's capabilities, opportunities and motivation towards PA both before and during COVID-19. Women were asked to indicate their level of agreement with various statements on a 5-point Likert scale. Table 4 shows the percentage of women who strongly agreed/agreed with each statement. Women's ability to be physically active dropped from 86.5% before COVID-19 to 58.6% during COVID-19. There was also a decline in women's reported opportunity to be physically active, with 88% agreeing they had the opportunity to be physically active before COVID, this dropped to 51.1% during COVID-19.

A logistic binary regression model was run to predict PA levels during COVID-19 from factors relating to the COM-B model; having the ability to be physically active, having opportunities to be physically active, feeling it is important to be physically active, finding exercise enjoyable and satisfying and feeling guilty when not exercising (Table 5). Women who agreed they had the opportunity to be physically active during COVID-19 were 5.7 times more likely to meet the PA guidelines than those who did not report having the opportunity to be physically active.

## 3.7 Sedentary behaviour

Almost eighty percent (79%) of the women survey reported their sedentary time increased during the COVID-19 pandemic. There was a statistically significant association between how

Table 5. Agreement with COM-B statements as predictors of meeting the PA guidelines during COVID-19.

| | | | | | | 95% CI for Exp (B) | |
| --- | --- | --- | --- | --- | --- | --- | --- |
| Predictor Variables | B | SE | df | Sig. | Exp (B) | Lower | Upper |
| Ability to be physically active | 1.132 | .383 | 1 | .003* | 3.101 | 1.464 | 6.566 |
| Opportunity to be physically active | 1.734 | .373 | 1 | .000* | 5.662 | 2.727 | 11.752 |
| Important to be physically active | 1.178 | .388 | 1 | .002* | 3.249 | 1.518 | 6.953 |
| Found exercise enjoyable and satisfying | 1.042 | .293 | 1 | .000* | 2.836 | 1.596 | 5.036 |
| Feel guilty when don't exercise | -.341 | .292 | 1 | .244 | .711 | .401 | 1.262 |

*Statistically significant result.

B- co-efficient for the constant.

SE-Standard error.

df-degrees of freedom.

Sig.-Significance.

Exp (B) Exponentiation of B coefficient (odds ratio).

CI- confidence intervals.

sedentary time had been affected by COVID-19 and meeting the PA guidelines ($X^2$ = 47.03, p = < .001). Those whose sedentary time had increased were less likely to meeting the PA guidelines than those whose sedentary time had decreased. There was no difference between change in sedentary time and awareness of the negative impacts of sedentary behaviour in pregnancy, however, there was a statistically significant association between those who agreed they knew how to exercise safely in pregnancy and decreased sedentary time (79.6% Vs. 20.4%, $X^2$ = 7.179, P = .007).

## 3.8 Brief Measure of Worry Severity (BMWS)

Women were asked eight questions on general worry; the scores were totalled to create a single worry score for each woman on a scale of 0–24. The mean score was 12.15 (SD 6.65). Based on the analysis from a one way ANOVA there was a significant difference in the scores for meeting the guidelines (M = 10.82, SD 6.48) and not meeting the guidelines (M = 12.59, SD 6.66) (F (1, 427) = 5.57, P = .018).

## 4. Discussion

This study explored how the restrictions resulting from the COVID-19 pandemic affected the physical activity and sedentary behaviour levels of pregnant women with gestational diabetes. It is clear from the results that the pandemic has vastly altered the activity levels of this group of women. The proportion of women meeting the physical activity guidelines dropped by 50% from before COVID-19 to during the pandemic. The decline seen here follows a similar pattern observed by other studies. Duncan and colleagues [39] found 42% of women decreased their PA levels during the pandemic, compared to 60% of the women in this study. The women in the study by Duncan and colleagues [39] were not pregnant and therefore it is likely that being pregnant and the increasing complexities which come with pregnancy may have been responsible for the higher level of decline in this study.

Based on the COM-B model, women's capability to be physically active dropped during the pandemic. This may have been due to activities such as swimming not being possible and women feeling they did not have the capability to be active in other ways. There was also a large drop in women's perceived opportunities for physical activity. This is likely to have been due to the closing of gyms, swimming pools and leisure centres, and the cancelling of face-to-face classes such as pregnancy yoga and Pilates. Some parks and green spaces were also shut during lockdown, further reducing opportunities for PA.

Sport England have consistently measured adult's capability, opportunities and motivation towards PA during the pandemic. During the first six weeks of lockdown, 67% of respondents agreed they had the opportunity to be physically active [36], compared to 51% in this study. The lower level of physical activity opportunities faced by the women in this study is likely to be due to the fact that pregnancy already limits physical activity opportunities with opportunities being further limited due to the pandemic. Findings from a logistic binary regression model showed having opportunities to be physically active was the largest predictor of meeting the PA guidelines during COVID-19. Therefore, opportunities need to be created for this group of women to prevent the decline seen here in future lockdowns. There is a need for the development of online PA/exercise classes, taken by an instructor qualified in prenatal exercise, in conjunction with health care professionals, for this group of women. Over three-quarters of the women in the study agreed they would take part in an online pregnancy exercise class if it was available.

Another factor which was associated with women's PA levels during COVID-19 was knowing how to exercise safely in pregnancy, with women who did not know how to exercise safely

in pregnancy being less likely to meet the PA guidelines during COVID-19. This highlights the importance of women being given suitable information and resources on how to exercise safely in pregnancy. Despite the NICE guidelines recommending pregnant women diagnosed with gestational diabetes are offered advice about changes in diet and exercise [30], the majority of the woman in the study felt it would be useful to be given information on the benefits of physical activity in pregnancy, especially with regards to gestational diabetes and that it would be useful to receive instruction on how and what types of PA to do in pregnancy. In Northern Ireland pregnant women are given 'The Pregnancy Book' at their booking appointment which contains two pages on physical activity in pregnancy; providing information on the PA guidelines in pregnancy (new addition in 2020 publication), the benefits of PA, exercises to avoid and some example of stretches and pelvic floor exercises [40]. It may be the case that women are inundated with information at their booking appointment and this information is not getting read or it is not in an accessible format for these women. It is clear that, although some information is being given to the women, a large percentage want more information, especially around gestational diabetes and instruction of how and what types of PA to do in pregnancy.

In this study, 42% of women had sought information to help them be physically active during lockdown. The most commonly reported places where information was sought were social media and websites; the least reported source was from health professionals. Research with midwives found that although midwives felt they were ideally placed to provide guidance and advice on PA in pregnancy, many believed they were not equipped and lacked knowledge and confidence to do so [41]. Midwives also raised the point that PA was only addressed at the booking appointment as a tick box exercise and it would only be discussed again if raised by the individual [41]. Midwives need to receive training and guidance around physical activity in pregnancy as they are ideally placed to help support women to become/remain active in pregnancy. However, they cannot be expected to take the full responsibility, it has been argued a whole systems approach is needed to normalise PA from preconception through to motherhood [42]. There has been good progress made on the availability of information on physical activity in pregnancy through the creation of the Active Pregnancy Foundation and the compiling of suitable content for physical activity in pregnancy through initiatives such as #thismummoves. However, it is likely that this information will be utilised by women who are active and physically literate. The real challenge is supporting women who are not currently active to become active. In addition, generally, the discussions around PA in pregnancy focus on uncomplicated pregnancies. There may also be further uncertainly and confusion around which types of physical activity are suitable and safe for women with gestational diabetes.

The change in maternity care, with a reduction in face-to-face appointments, may have also had an impact on women's physical activity levels. Exposure to health literature in hospitals and waiting rooms would have been reduced, as would the opportunity for general conversations around topics such as PA. With these changes set to remain for some time, changes need to be made to allow for alternative opportunities to address PA.

Those with an undergraduate or higher degree were more likely to meet the PA guidelines during COVID-19, although there was no association between education and meeting the PA guidelines pre COVID-19. It may be that those with a degree are more likely to have greater access to online information and more space inside their homes to exercise when other options were not available. Therefore, the pandemic may have had a disproportionate effect on those with lower means, highlighting the importance of providing women with information on not only how to exercise safely in pregnancy but also on how to do this with limited space and equipment.

It cannot be ignored that the most commonly reported reason for decline in physical activity levels was fear of leaving the house due to COVID-19. There are a number of factors which

may have affected women's fear levels. Firstly, on the 16th March pregnant women were placed in the vulnerable category, potentially increasing worry and anxiety for this group of women. Pregnancy is already known as a time of higher anxiety levels [43] and there was a lot of uncertainty over whether or not pregnant women were at higher risk of getting COVID-19 and in turn becoming more seriously affected by the virus. In addition to pregnancy, this group of women were also managing the complex health condition of GDM. The prevalence of anxiety scores of women with GDM is estimated to be approximately 29.5% [44]. Together with the finding that in the general population, the COVID-19 pandemic has been found to have a negative impact on mental health, with 36.8% prevalence of poor mental health, compared to none pandemic levels of 25% [45], this indicates that the anxiety scores of women with GDM are likely to be higher than 29.5%. An online exercise programme would remove the need for woman to leave their home, which is particularly important for woman who experience fear of leaving the house as a barrier to physical activity. In addition, PA is known to help lower anxiety levels [46].

The BMWS scale provides a broad assessment of worry, examining various functions of dysfunctional worry such as impairment and interference, uncontrollability and mood disturbance [47]. Evidence suggests the BWMS scale is a reliable measure of antenatal anxiety and has been found to be a good predictor of postnatal depression [48]. The mean worry score in this study was 12.15 (SD 6.65). Austin and colleagues defined dysfunctional worry as scores over 12 on the BMWS scale [47]. Almost half (45%) of the women in our study had a BMWS over 12, indicating high levels of dysfunctional worry. In comparison, in a study of pregnant women in New Zealand, approximately 14% of the sample experienced dysfunctional worry as measured by the BMWS scale [49]. It is likely that both the fact that the women in our study had GDM and were pregnant during a pandemic resulted in higher worry scores. Women with higher worry scores were more likely to attribute the decline in their PA to finding lockdown hard, lack of energy and lack of motivation. Lack of energy and lack of motivation are both associated with depressive feelings [50]. As higher BMWS scores have been found to be predictors of postnatal depression it is important that this group of women receive appropriate support in the postnatal period as it would suggest these women are at greater risk. During future waves of COVID-19 and future pandemics pregnant women with GDM need reassurance and appropriate advice and support to try to reduce the levels of worry experienced.

There was a statistically significant difference between physical activity levels of the low worry group (0–12 BMWS score) and the high worry group (13–24 BMWS score). PA has been found to reduce depression and anxiety [46]. However, in this study it was not possible to say whether those whose worry scores were lower was a result of the PA or if the those in the group with higher worry scores faced more barriers to physical activity, such as lack of motivation and, therefore, had lower PA levels; further research is needed to investigate this.

Finally, it cannot be ignored that almost 20% of women in the study increased their PA levels during the COVID-19 pandemic. This is a positive finding with and one of the most commonly cited reasons being women felt it was important to be active due to GDM. This highlights the importance of giving information about the benefits of PA, especially in relation to GDM.

## 5. Conclusions

The COVID-19 pandemic has had a considerable impact on the activity levels of pregnant women with gestational diabetes and highlights the need for targeted public health initiatives. Women with GDM need to be educated on the safety and benefits of engaging in PA and ways to do this from their own homes if fear of leaving the house due to COVID-19 is a barrier for

them. An online exercise class developed for women with GDM and delivered by an exercise instructor qualified in prenatal exercise may be one possible solution. This group of women also need extra support to reduce fears and worry around the COVID-19 pandemic as they are currently experiencing high levels of worry which may lead to higher levels of postnatal depression. Restrictions and changes to daily life due to COVID-19 are likely to be around for some time and therefore it is important that changes and adaptations are made to avoid long term health impacts due to reduced PA levels and increased worry. The findings from this study will be useful during future waves of COVID-19 and other pandemics.

## 6. Strengths and limitations

To the best of our knowledge this is the first study investigating the impact of COVID-19 on the physical activity and sedentary behaviour levels of pregnant women with gestational diabetes. This study has useful findings which could prove helpful in future lockdowns and pandemics.

Limitations of the study include the self-reporting of physical activity levels resulting in potential over-reporting and reliance on women's recall of their physical activity levels pre-COVID-19 may have resulted in inaccuracies. However, given the situation it was felt a survey was the best method to use for data collection. As the women self-reported their PA levels both pre COVID-19 and during COVID-19 it is likely that reporting inaccuracies are non-differential, therefore, not affecting the general level of decline that has been seen.

Secondly, the sample may not be representative of the UK as a whole as respondents had a higher level of education, with 59% having an undergraduate degree or higher degree. Although there is some disagreement over the percentage of the UK population who hold a degree, it is estimated to be approximately 27% [51]. Also, a higher percentage of respondents selected white as their ethnicity than the UK average (93% Vs 86%). This may have had some impact on the results as people from Black, Asian and minority ethnic (BAME) backgrounds are at higher risk for GDM and have been found to be at greater risk of developing severe illness if they contract COVID-19. Therefore, women in this group may have had a difference experience of the COVID-19 pandemic and are underrepresented in this study. One of the limitations to recruiting online is selection bias, with individuals from BAME and lower income backgrounds being less likely to be represented. Atkinson and colleagues have called for long-term strategies to build relationships with hard to reach groups [3].

## 7. Recommendations

- Midwives need to be offered additional evidence-informed training and guidance on motivational strategies, to enhance pregnant women's potential to become and remain active during pregnancy.

- Evidence based guidance is needed from the Royal College of Midwives/Royal College of Obstetricians & Gynaecologists with reference to PA in groups with complex health conditions such as GDM.

- Pregnant women need to be educated on the health benefits of PA in pregnancy, especially around GDM and need to receive instruction on how and what types of PA to do in pregnancy.

- Opportunities need to be created for this group of women to prevent the decline seen here in future lockdowns and pandemics. There is the need for the development of online PA classes, taken by an instructor qualified in prenatal exercise, in conjunction with health care

professionals, for this group of women. Additionally, resources are needed on how to be active at home, with limited space and equipment.

## Acknowledgments

We would like to thank all the women who took part in the survey.

## Author Contributions

**Conceptualization:** Medbh Hillyard, Marlene Sinclair, Marie Murphy, Karen Casson, Ciara Mulligan.

**Data curation:** Medbh Hillyard.

**Formal analysis:** Medbh Hillyard.

**Investigation:** Medbh Hillyard, Marlene Sinclair, Marie Murphy, Karen Casson, Ciara Mulligan.

**Methodology:** Medbh Hillyard, Marlene Sinclair, Marie Murphy, Karen Casson, Ciara Mulligan.

**Project administration:** Medbh Hillyard, Marlene Sinclair, Ciara Mulligan.

**Supervision:** Marlene Sinclair, Marie Murphy, Karen Casson, Ciara Mulligan.

**Visualization:** Medbh Hillyard.

**Writing – original draft:** Medbh Hillyard.

**Writing – review & editing:** Medbh Hillyard, Marlene Sinclair, Marie Murphy, Karen Casson, Ciara Mulligan.

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
