## [Decision Letter · Decision Letter 0]

13 Jan 2021

PONE-D-20-35725

The impact of COVID-19 on the physical activity and sedentary behaviour levels of pregnant women with gestational diabetes.

PLOS ONE

Dear Dr. Hillyard,

Thank you for submitting your manuscript to PLOS ONE. After careful consideration, we feel that it has merit but does not fully meet PLOS ONE’s publication criteria as it currently stands. Therefore, we invite you to submit a revised version of the manuscript that addresses the points raised during the review process.

Please respond to the reviewer comments and to my queries in your response. 

We look forward to receiving your revised manuscript.

Kind regards,

Edward Mullins

Academic Editor

PLOS ONE

Journal Requirements:

2. In your Methods section, please include additional information about your dataset and ensure that you have included a statement specifying whether the collection method complied with the terms and conditions for the website.

Additional Editor Comments:

Thank you for this article, addressing an important issue in a group of women whose pregnancies are more complicated and in whom PA can make a great differenc in glycaemia and potentially perinatal outcomes.

Please note the reviewer comments and address these.

Additional comments:

Please edit and focus the introduction which is rather long at present.

Educational qualifications - suggest amend to indicate highest qualification gained as the results are not interpretable at present.

Note that reason for increased exercise during pandemic was largely attributed to nice weather - now in winter, we are likely to be seeing lower levels of outdoor and overall PA in this group during lockdown.

How would you advise researchers to address the issue of responder bias and the difficulty in getting an ethnically representative sample for this sort of study in future?

Reviewers' comments:

Reviewer's Responses to Questions

**Comments to the Author**

1. Is the manuscript technically sound, and do the data support the conclusions?

Reviewer #1: Yes

Reviewer #2: Partly

2. Has the statistical analysis been performed appropriately and rigorously? 

Reviewer #1: No

Reviewer #2: Yes

3. Have the authors made all data underlying the findings in their manuscript fully available?

Reviewer #1: No

Reviewer #2: No

4. Is the manuscript presented in an intelligible fashion and written in standard English?

Reviewer #1: Yes

Reviewer #2: Yes

5. Review Comments to the Author

Reviewer #1: Many thanks for asking me to review this qualitative analysis examining the impact of Covid-19 lockdowns on the physical activity behaviour of women whose pregnancies are complicated by gestational diabetes.

While this represents an important piece of work further and adds to the public health impact that lockdowns have, there are comments that need to be addressed prior to considering publication.

Major comments:

1. Tables. In general they are difficult to follow. More explanation needs to be provided as to the parameters included in the tables and relevance of statistical analyses needs to be considered:

i) Table 1: the column headings are misleading – I would suggest maternal demographics rather than age for the first column. Why are both range of age and mean values included? Are the data normally distributed?

The inclusion of the p value for level of qualification suggests that a chi squared table for all categories has been generated? Was a bonferroni correction applied? It may be better to consider calculating the p value on two simple categorical alone eg qualifications vs no qualifications, and/or higher degree versus no higher degree as the number of variables included in the current analysis renders a high probability that the statistical significance could be related to chance. The same comment applies to table 2 and treatment for GDM.

i) Tables 3 and 5: these tables are confusing and the headings need explanation

2. Lines 276-280: The mean gestational age of women in the sample is 22.1 weeks. The mean GA that women were diagnosed with GDM is 24.1 weeks. Do the authors have a comment on this discrepancy? The mean age does suggest that a significant proportion did not have an index pregnancy complicated by GDM given the time frame at which NICE recommends testing for this. Were women with T1DM/ T2DM perhaps inadvertently included?

3. Discussion: caution needs to be exercised in terms of over-stating the results. While the results indicate that women’s levels of activities dropped during covid, these are questionnaires asking women about past behaviours i.e. recall bias needs to be considered. Furthermore, at what point were women taking “pre-covid times”? Would this potentially not have included much earlier gestational age when physical activity would have been much less limited?

Minor comments:

1. Lines 77-79 are not clear: do the authors mean “reported cases of severe covid”?

2. Line 113: Statement is narrow. Women of other non-white European ethnic groups such as Arab/ South East Asian/ Hispanic/ Black African Caribbean are also at risk of developing gestational diabetes.

3. Lines 139-141: evidence already explained earlier in the text therefore does not need to be repeated.

4. Power calculation for sample size: what outcome was the power calculation based on and what statistical analysis was used to reach the sample size?

5. Lines 297- 308: are all the questions that women were asked included in this section? Would it perhaps be better to include the actual questionnaire for clarity? If not feasible within the main text, perhaps in the supplementary information?

6 Line 292: This statement is confusing. Do the authors mean that x proportion of women adhered to PA guidelines according to the results of the questionnaire? Or is the statement inferred from the mean number of days that women exercised for over 30 minutes? If the latter, how they arrived at the calculation needs to be explained.

7. Line 376/377. This is not necessary to include. If women say they are more sedentary it would follow that they are likely to be less physically active

8. Lines 340-343. Proportions here are of more value than reporting “x

Reviewer #2: It is a relevant research that investigates the relationship between the level of physical activity in pregnant women during the pandemic. It identifies a trend towards a reduction in physical activity levels during the pandemic and the lockdown enforced by the Government among pregnant women with Gestational diabetes.

Physical activity tends to reduce for pregnant women as pregnancy progresses due to the physiological changes of pregnancy. How this observation affected their results was not clearly addressed in either the introduction or the discussion sections of the manuscript.

The introduction is a bit too expansive and can be modified to become more concise based on the stated aims and objectives of the research.

The research design and statistical analysis is appropriate. Also, the use of online platforms to share the research questionnaire is acceptable due to concerns about the pandemic.

6. PLOS authors have the option to publish the peer review history of their article (what does this mean?). If published, this will include your full peer review and any attached files.

Reviewer #1: No

Reviewer #2: No

---

## [Author Response · Author response to Decision Letter 0]

19 May 2021

RE: “The impact of COVID-19 on the physical activity and sedentary behaviour levels of pregnant women with gestational diabetes.” Response to Reviewers 27th April 2021. 

Reviewer’s comments Response 

Please edit and focus the introduction which is rather long at present Introduction has been edited and reduced in length 

Educational qualifications - suggest amend to indicate highest qualification gained as the results are not interpretable at present.

 Changed to degree/no degree 

How would you advise researchers to address the issue of responder bias and the difficulty in getting an ethnically representative sample for this sort of study in future?

Suggestions added Section 6: strengths and limitations

 Table 1: the column headings are misleading – I would suggest maternal demographics rather than age for the first column. Why are both range of age and mean values included? Are the data normally distributed?

Table simplified. Maternal characteristics heading added to first column. Range removed. 

It may be better to consider calculating the p value on two simple categorical alone eg qualifications vs no qualifications, and/or higher degree versus no higher degree as the number of variables included in the current analysis renders a high probability that the statistical significance could be related to chance. The same comment applies to table 2 and treatment for GDM.

 Table simplified. Changed to degree/no degree. P value given between the two categories. 

Table 2 GDM management changed to diet/medication 

Tables 3 and 5: these tables are confusing and the headings need explanation

Tables simplified and headings for tables 3 and 5 explained 

Lines 276-280: The mean gestational age of women in the sample is 22.1 weeks. The mean GA that women were diagnosed with GDM is 24.1 weeks. Do the authors have a comment on this discrepancy?

 In line with the NICE guidelines (NG3) women who have previously had GDM should be offered early self monitoring of blood glucose or a 75g 2h Oral Glucose Tolerance Test as soon after booking as possible. Therefore these results suggest a larger number of women in the sample had GDM in a previous pregnancy and were therefore diagnosed earlier than the 24-28 week standard OGTT. This is in keeping with the fact that 62% of the sample were multiparous.

Discussion: caution needs to be exercised in terms of over-stating the results. While the results indicate that women’s levels of activities dropped during covid, these are questionnaires asking women about past behaviours i.e. recall bias needs to be considered. Furthermore, at what point were women taking “pre-covid times”? Would this potentially not have included much earlier gestational age when physical activity would have been much less limited?

 It has been highlighted in section 6 Strengths and limitations that self-reporting and reliance on recall can result in inaccuracies. In addition, it has been highlighted that as women were self-reporting activity levels at both time points reporting inaccuracies are likely to be non-differential and therefore not affecting the general level of decline seen.

In addition, the PA levels by trimester have been added to section 3.2 physical activity levels, showing that PA levels were lower in T1 and increased in T2.

Lines 77-79 are not clear: do the authors mean “reported cases of severe covid”?

 Removed

 Line 113: Statement is narrow. Women of other non-white European ethnic groups such as Arab/ South East Asian/ Hispanic/ Black African Caribbean are also at risk of developing gestational diabetes.

 Line 149-150 made broader 

Lines 139-141: evidence already explained earlier in the text therefore does not need to be repeated.

 Earlier evidence/explanation removed to avoid repetition 

Power calculation for sample size: what outcome was the power calculation based on and what statistical analysis was used to reach the sample size?

 The outcome the power calculation was based on was the number of women diagnosed with GDM. There were 731,213 births in the UK in 2018. It is estimated that approximately 4.4% of women in the UK have GDM. This would mean approximately 32,173 women are diagnosed with GDM annually in the UK. Based on a power calculation with a population size of 32,173, at a 95% confidence level and 5% error, it was calculated that 380 women would need to complete the survey to provide representative results. 

Lines 297- 308: are all the questions that women were asked included in this section?

 No this is not all the questions which were asked but the questions relevant to this analysis.

 Line 292: This statement is confusing. Do the authors mean that x proportion of women adhered to PA guidelines according to the results of the questionnaire? Or is the statement inferred from the mean number of days that women exercised for over 30 minutes? If the latter, how they arrived at the calculation needs to be explained.

 Further explanation to how the percentage was arrived at is given. 

Line 376/377. This is not necessary to include. If women say they are more sedentary it would follow that they are likely to be less physically active

 Sedentary time and PA are two different behaviours. An individual can have increased their sedentary time but also have increased with PA, therefore looking at both of these behaviours separately is important.

 Lines 340-343. Proportions here are of more value than reporting “x

 Proportions are given in the original manuscript 

Physical activity tends to reduce for pregnant women as pregnancy progresses due to the physiological changes of pregnancy. How this observation affected their results was not clearly addressed in either the introduction or the discussion sections of the manuscript.

 Information on PA levels during each trimester has been included in section 3.2

The introduction is a bit too expansive and can be modified to become more concise based on the stated aims and objectives of the research.

 Introduction has been edited and made more concise.

---

## [Decision Letter · Decision Letter 1]

28 Jun 2021

The impact of COVID-19 on the physical activity and sedentary behaviour levels of pregnant women with gestational diabetes.

PONE-D-20-35725R1

Dear Dr. Sinclair,

We’re pleased to inform you that your manuscript has been judged scientifically suitable for publication and will be formally accepted for publication once it meets all outstanding technical requirements.

Kind regards,

Edward Mullins

Academic Editor

PLOS ONE

Additional Editor Comments (optional):

Many thanks for revising your manuscript. I think this will be a useful contribution to care for women with GDM.

Reviewers' comments:

Reviewer's Responses to Questions

**Comments to the Author**

1. If the authors have adequately addressed your comments raised in a previous round of review and you feel that this manuscript is now acceptable for publication, you may indicate that here to bypass the “Comments to the Author” section, enter your conflict of interest statement in the “Confidential to Editor” section, and submit your "Accept" recommendation.

Reviewer #2: All comments have been addressed

2. Is the manuscript technically sound, and do the data support the conclusions?

Reviewer #2: Yes

3. Has the statistical analysis been performed appropriately and rigorously? 

Reviewer #2: Yes

4. Have the authors made all data underlying the findings in their manuscript fully available?

Reviewer #2: Yes

5. Is the manuscript presented in an intelligible fashion and written in standard English?

Reviewer #2: Yes

6. Review Comments to the Author

Reviewer #2: I have no further comments. The queries I highlighted in my previous review have been addressed in this revised manuscript.

The introduction has been reduced and is more focused in its present form. I note also the corrections made under the results section including information comparing PA levels during each trimester has been included in section 3.2, Also, a statement that the full questionnaire and data sets are available online has been added.

7. PLOS authors have the option to publish the peer review history of their article (what does this mean?). If published, this will include your full peer review and any attached files.

Reviewer #2: **Yes: **Dr Michael Chudi Ezeanochie

---

## [Editor Report · Acceptance letter]

13 Aug 2021

PONE-D-20-35725R1 

The impact of COVID-19 on the physical activity and sedentary behaviour levels of pregnant women with gestational diabetes. 

Dear Dr. Sinclair:

I'm pleased to inform you that your manuscript has been deemed suitable for publication in PLOS ONE. Congratulations! Your manuscript is now with our production department. 

Kind regards, 

on behalf of

Dr. Edward Mullins 

Academic Editor

PLOS ONE